# Reward System EEG–fMRI-Pattern Neurofeedback for Major Depressive Disorder with Anhedonia: A Multicenter Pilot Study

**DOI:** 10.3390/brainsci15050476

**Published:** 2025-04-29

**Authors:** Daniela Amital, Raz Gross, Nadav Goldental, Eyal Fruchter, Haya Yaron-Wachtel, Aron Tendler, Yaki Stern, Lisa Deutsch, Jeffrey D. Voigt, Talma Hendler, Tal Harmelech, Neomi Singer, Haggai Sharon

**Affiliations:** 1Barzilai Medical Center, Ashkelon 7830604, Israel; danielaam@bmc.gov.il; 2Sheba Medical Center, Ramat Gan 52621, Israel or razg@post.tau.ac.il (R.G.); nadav.goldental@sheba.health.gov.il (N.G.); 3Department of Epidemiology and Preventive Medicine & Department of Psychiatry, School of Public Health & School of Medicine, Faculty of Medical and Health Sciences, Tel Aviv University, Tel Aviv 6997801, Israel; 4ICAR Collective and Brus Rappaport Medical Facility of the Technicon, Haifa 3525433, Israel; eyal.fruchter@psmh.health.gov.il (E.F.); h_yaron@rambam.health.gov.il (H.Y.-W.); 5GrayMatters Health, Haifa 3303403, Israel; yaki@graymatters-health.com (Y.S.); or thendler@tauex.tau.ac.il (T.H.); talharmelech@gmail.com (T.H.); 6Biostats Statistical Consulting Ltd., Modiin 7170717, Israel; lisa@ldbiostats.com; 7Medical Device Consultants of Ridgewood, Ridgewood, NJ 07450, USA; meddevconsultant@aol.com; 8Department of Psychology, Tel Aviv University, Tel Aviv 6997801, Israel; 9Sagol Brain Institute, Tel Aviv Sourasky Medical Center, Tel Aviv 6997801, Israel; singer.neomi@gmail.com (N.S.); haggaisharon@gmail.com (H.S.); 10Faculty of Medical and Health Sciences, Tel Aviv University, Tel Aviv 6997801, Israel; 11Institute of Pain Medicine, Department of Anesthesiology and Critical Care, Tel Aviv Sourasky Medical Center, Tel Aviv 6423906, Israel

**Keywords:** EFP biomarker, neurofeedback, self-neuromodulation, depression, anhedonia, depression treatment, reward system, ventral striatum, biomarker, personalized treatment

## Abstract

**Background/Objectives:** Up to 75% of patients with major depressive disorder (MDD) exhibit persistent anhedonia symptoms related to abnormalities in the positive valence system. Cumulative evidence points to brain dysfunction in the reward system (RS), including in the ventral striatum, in patients with MDD with anhedonia. This study aims to evaluate the safety and efficacy of a novel neurofeedback (NF) device (termed Prism) which incorporates the EEG–FRI-Pattern biomarker of the reward system (RS-EFP) for use in self-neuromodulation training (RS-EFP-NF) for alleviating depression in patients with MDD with anhedonia. **Methods:** A total of 49 adults (age range: M = 39.9 ± 11.03) with a DSM-5 diagnosis of MDD with anhedonia (per a SHAPS-C score ≥ 25) were screened for the administration of ten sessions of RS-EFP-NF twice a week on nonconsecutive days. Depression and anhedonia severity was assessed, respectively, by HDRS-17 and SHAPS-C at baseline, midway, and treatment end. **Results:** A total of 34 patients (77%) completed the protocol and were included in the analyses. No device-related adverse events were serious or required treatment. Depression symptoms were reduced at end of treatment as indicated by the HDRS-17, with a reduction of eight points on average (95% CI: −10.5 to −5.41, *p* < 0.0001), a clinical improvement rate of 78.47%, and a remission rate of 32.25%. Anhedonia, as indicated by the SHAPS-C score, was diminished, showing an average reduction of 6.3 points (95% CI: −8.51 to −4.14, *p* < 0.0001). **Conclusions:** Self-neuromodulation using RS-EFP-NF is a promising and safe treatment for MDD with anhedonia. The intervention demonstrates substantial clinical effects on both depression and anhedonia symptoms, with high patient acceptability and retention. Prism may address a critical mechanism-driven treatment gap for anhedonia that often persists despite conventional therapies. Larger controlled implementation, efficacy, and dosing studies are warranted.

## 1. Introduction

The diagnosis of major depressive disorder (MDD) requires at least five depressive symptoms, one of which may be anhedonia—the loss of pleasure or interest in daily activities perceived as rewarding during a premorbid state [1]. Anhedonia can be present in up to 70% of MDD diagnoses [1,2]. With a prevalence of 14.5 million individuals with MDD, approximately 10.3 million adults with MDD would be considered to have anhedonic depression. Importantly, anhedonia is positively related to the severity of MDD [3], prolonged disease course [4], worse long-term prognosis [5], and a higher suicide rate [1,6]. Anhedonia has been found to be a predictor of poor outcomes in patients taking antidepressant medications, particularly serotonin uptake inhibitors (SSRIs) [7,8,9], as these medications fail to address reward-related symptoms associated with anhedonia [10,11]. There is a clinical need for therapies specifically designed to treat anhedonia-related MDD [12].

As shown consistently across human imaging studies, depression, in general, and anhedonic symptoms, specifically, are associated with deficient activation of the ventral striatum (VS), a core aspect of the subcortical reward circuit [13,14,15]. This relationship has been further characterized by Pizzagalli (2014), who described how dysfunctional reward processing manifests neurobiologically as anhedonia [16]. Decades of experimental research in animals and humans have emphasized the role of the meso–cortico–limbic circuit in different sub-domains of reward system processes, including anticipation and consumption of rewards, valuation, and decision-making concerning efforts to pursue a possible reward, as well as learning from this for future occurrences [17,18,19]. Many of these studies refer to the role of neuroanatomical connections and dopamine secretion in the meso–limbic circuit [20]. Although other nondopaminergic mechanisms also play roles in reward pleasure (i.e., “liking”), dopamine has received attention because of its critical effects on reward anticipation and incentive (i.e., “wanting”) of pleasurable stimuli [21]. Therefore, anhedonia is reflected both at the molecular level, involving deficits in dopamine production, and at the structural level, indicating impaired processing within meso–cortico–limbic structures [9].

Despite the high prevalence of anhedonia in MDD and its association with worse outcomes, few interventions specifically target it. The growing neurobiological understanding of its mechanism supports the idea that interventions targeting the reward system could be beneficial for MDD with anhedonia. Prior studies using behavioral activation techniques and pharmacological agents directed at reward-related neural circuits have shown notable improvements in depressive symptoms by mitigating anhedonia [10,22,23].

A recently developed, innovative form of neurofeedback (NF), inspired by [24,25] and named Prism for Depression™ (Gray Matters Health, Haifa, Israel), follows the rational of probing reward system activation (RS) for the treatment of depression with anhedonia. Uniquely, this approach targets the relevant neural mechanism by analytically integrating simultaneous EEG and fMRI recordings to create a group-level biomarker for VS activation during reward processing. By utilizing machine learning for estimating fMRI activity in the VS from the simultaneously acquired EEG data, a set of coefficients (named Reward System EEG–fMRI-Pattern (RS-EFP)) are derived. This EFP serves as an fMRI-informed measure of EEG signals related to reward system activity during the processing of pleasurable stimuli [24]. An earlier implementation of the RS-EFP was validated in healthy individuals by Singer et al. [25], demonstrating its specificity for detecting reward system activation particularly in the ventral striatum. The validation studies demonstrated that this model predicts activation in the VS and additional functionally relevant reward-related regions (such as the insula and anterior cingulate cortex [ACC]) to a greater extent than EFP models derived from other anatomical regions of the brain [25]. The Prism for Depression™ approach re-implements the RS-EFP biomarker as the self-neuromodulation target during EEG-NF, emphasizing the active role patients play in modulating their own neural activity in the reward system. Unlike conventional NF approaches that target general EEG patterns, the RS-EFP-NF approach uniquely provides anatomically specific feedback derived from the representation of deep brain activity in reward-related structures, potentially offering more precise neuromodulation of anhedonia-related neural circuits [26].

The goal of this study was to prospectively evaluate the safety and clinical efficacy of RS-EFP-NF integrated in a dedicated device and audio–visual feedback interface (i.e., Prism for Depression™, Figure 1 and Figure 2) in patients with MDD with anhedonia.

## 2. Methods

### 2.1. Participants and Recruitment

The inclusion criteria were as follows:Primary diagnosis of MDD with anhedonia, with a SHAPS-C score ≥ 25;MDD diagnosis determined via the neuropsychiatric interview (MINI for DSM-5);Aged 22 to 50;Any gender;High school diploma or equivalent;Right-handed if undergoing an MRI [27];Normal or corrected-to-normal vision and hearing;Ability to provide signed, informed consent;Ability to adhere to the study schedule;Concomitant psychotropic medications allowed if they are at a stable dose for 4 weeks prior to the study.

The exclusion criteria were as follows:History of psychotic disorder or bipolar I;Moderate or severe substance use disorder within 3 months of screening;Lifetime diagnosis of autism or intellectual disability allowed at the investigators’ discretion;Benzodiazepines that cannot be ceased for the duration of the study (with a washout period of at least 2 weeks prior to the first Prism training session) or that cannot be replaced with short-acting benzodiazepines;Current diagnosis of PTSD;Treatment-resistant depression, defined as episodes that did not have a ≥50% symptom reduction to at least two full trials;Any past or current use of DA (dopamine agonist)-acting drugs;Recent initiation of any evidence-based MDD psychotherapy was excluded but continuation of established therapy was allowed.

Participants were recruited from four medical centers in Israel (Ramban Medical Center, Sheba Medical Center, Sourasky Medical Center, and Barzilai Medical Center). The fMRIs were conducted at Sourasky Medical Center, and the NF only at the other three medical centers. The research protocol was approved by the ethics committee of each participating clinical site. The study was registered on the Israel Ministry of Health website (MOH_2022-02-22_010631). The study took place from 22 February 2023 to 6 March 2024. A planned sample size of 30 was chosen for patients with MDD with anhedonia based on the rule of thumb recommendation [28].

The Consensus on the Reporting of Experimental Design of clinical and cognitive–behavioral NF studies (CRED-nf) best practices checklist was used (Appendix A).

### 2.2. Study Design, Device Description, Outcome Measures, and Statistical Methods

#### 2.2.1. Study Design

The study was a prospective, single arm, open-label treatment trial with Prism for Depression™ aimed at assessing its safety and efficacy in patients with MDD with anhedonia. The goal was to train individuals to upregulate their RS-EFP biomarker, (i.e., self-neuromodulation) while probing various sub-domain of reward processing. This study consisted of subjects participating in the following visits: screening and eligibility confirmation; baseline clinical assessment; 10 Prism sessions; mid-study clinical assessment; and post-Prism clinical assessments. The study also included two optional fMRI scans (before and after the 10 sessions), which are not included in the current report.

#### 2.2.2. Device Description

Using the RS-EFP biomarker as a modulation target in NF training, Prism aims to probe different components of reward processing related to depression and anhedonia, including reward anticipation and consumption. The RS-EFP calculation involves processing EEG data to extract the time–frequency features that best predict VS BOLD activation during such reward processing. The real-time EEG signal was decomposed into relevant frequency bands, and the resulting features are weighted according to the model coefficients to provide a continuous estimate of VS-related activity. The RS-EFP biomarker in the Prism device is operationalized through 6 specific EEG electrodes (CZ, PZ, C3, C4, P3, and P4 (referenced as FCz)) that record brain activity patterns most predictive of VS activation [25].

During the Prism NF training session, the patient was instructed to control an interactive audio–visual scenario presented on a screen (based on the principles presented in [25,29]). During each of the 10 training sessions, the participant was seated in front of a monitor, wearing a wireless EEG headset and watching an interactive scene with avatars. The participant was instructed to find an effective mental strategy—such as a memory of an experience, a song, or other sensation that evokes a happy, excited, or satisfied emotion—and induces changes in the scenario’s interface that correspond with successful up-modulation of the online computed RS-EFP biomarker. The patient was informed about the success in up-modulation the RS-EFP by changes in the audio–visual feedback interface as described in Figure 1 and Figure 2.

The Prism NF training protocol specifically emphasizes self-neuromodulation, where patients actively learn to control their brain activity through dedicated NF interface aimed to activate reward-related processes. As the participant engages in various mental strategies, Prism records an EEG from 6 electrodes and computes the RS-EFP biomarker every second. As described in Figure 1a, when the biomarker level goes up, the avatar gradually goes for a walk with the pet, providing a real-time representation of the patient’s RS-related brain activity and teaching control in an evolutional and repetitive way. The session’s structure and the scenario were designed to follow the main sub-domains for reward processing which are the reward anticipation and consumption (corresponding to “wanting” and “liking”).

Each Prism session has 5 cycles (3 min each), where each cycle is composed of three repeated blocks (Figure 1b). In the first stage of each block, the patient watches an animation of a dog trying to get its owner’s attention, creating anticipation for what will happen next (anticipation processing stage, 25 s). Self-neuromodulation success at this stage of anticipation results in the scenario moving to the next level, where the owner is taking the pet for an outside walk and an auditory–visual reward is presented (assumed to be a rewarding feedback for incentive-/wanting-related processing, 10 s). Then, if the feedback system detects that the RS-EFP is increasing in response to the reward, the patient will obtain an additional auditory–visual reward (feedback for reward consumption, 20 s). At this point, the patient is asked to maintain the level of the neural response to reward consumption. If they succeed, they will obtain an additional reward (brain state holding feedback;). Processing during this time is assumed to be a mixture reflecting incentive and consumption. For additional details on feedback interface changing stages, please see Figure 1a.

#### 2.2.3. Outcome Measures

The primary outcome measure was the Hamilton Depression Rating Scale (HDRS-17) change from baseline. Clinically meaningful improvement was defined as a 4–6-point reduction, and clinically substantial improvement was defined as a 7–12-point reduction [30]. The following hypotheses were tested for the primary efficacy endpoint:
Null Hypothesis: mean change from baseline to the post-Prism training visit in the Clinician HDRS-17 (week 6), HDRS ≥ −4;Alternative Hypothesis: mean change from baseline to the post-Prism training visit in HDRS-17 (week 6), HDRS < −4.

Other endpoints for HDRS included the following: proportion of responders (at least 50% reduction from baseline in the total HDRS score) and proportion of remitters (total HDRS score of 7 or lower at the post-NF training visit).

The secondary efficacy endpoints included measuring the change in anhedonia from baseline to the post-NF training visit via the clinician-administered Snaith–Hamilton Pleasure Scale (SHAPS-C); the Clinical Global Impression-Improvement (CGI-I) score at each visit until the post-NF training visit; the change from baseline to the post-NF training visit in the Quick Inventory Depressive Symptomatology (QIDS-SR-16); the change from baseline to the post-NF training visit in the General Anxiety Disorder-7 (GAD-7) and the change from baseline to the post-NF training visit in the Patient Health Questionnaire (PHQ-9).

Safety endpoints included the frequency, severity, and causality of adverse events (AEs) related and unrelated to Prism NF training.

#### 2.2.4. Statistical Methods

Statistical analyses were performed using SAS^®^V9.4 (SAS Institute, Cary, NC, USA). Three analysis sets were defined as follows: (1) Full Analysis (FA) set (all enrolled subjects with ≥1 Prism session); (2) Efficacy Analysis (EF) set (all FA subjects meeting inclusion criteria and completing 10 sessions); and (3) Per Protocol (PP) analysis set (all EF subjects without major protocol violations and with 10 completed sessions with detectable signal). The results are presented for the EF set, as no significant differences were found among the analysis sets.

The primary endpoint (change in the HDRS-17 from baseline to post-NF training) was analyzed using a linear mixed model for repeated measures. The model included visit as a categorical factor (Mid-NF and Post-NF), baseline HDRS-17 as a covariate, and site as a random effect (with an unstructured covariance matrix to account for within-subject correlations). If the model did not converge with site as a random effect, it was entered as a fixed effect. The null hypothesis (mean HDRS change ≥ −4) was tested against the alternative hypothesis (mean HDRS change < −4), with the rejection criterion being the upper bound of the 95% CI below −4.

For the secondary endpoints (SHAPS-C, CGI-I, QIDS-SR-16, GAD-7, and PHQ-9), similar mixed models were employed. For the SHAPS-C and CGI-I, repeated measures models were used with respective baseline values as covariates (except for CGI-I, which has no baseline). For the remaining measures, ANCOVA models were used with the baseline values as the covariates. A hierarchy approach was adopted for the primary and secondary endpoints to control type I errors due to multiple endpoint testing. Thus, the primary endpoint was first analyzed and only if successful would the secondary endpoints be analyzed.

Site poolability was assessed by adding site to the primary endpoint analysis model and testing at a 10% significance threshold. The response was defined as ≥ 50% reduction in HDRS-17 scores and remission as HDRS-17 score ≤ 7 at the study’s end. For comparison of the means (i.e., continuous variables), the two-sample *t*-test or the Wilcoxon rank-sum test was used as appropriate. For a comparison of the proportions (i.e., categorical variables), the chi-squared test or Fisher’s exact test was used as appropriate.

## 3. Results

The disposition of the participants is detailed in Table 1. Of 49 screened participants, 44 were enrolled (FA). Three withdrew prior to any sessions, and seven withdrew before the final assessment (23% attrition), resulting in thirty-four participants in the EF/PP sets. There was no statistically significant difference among the sites for the primary endpoint, allowing for data pooling.

### 3.1. Safety Analysis (AEs)

While 25% of the subjects experienced AEs (20 AEs in total), the majority were mild in nature with 12 mild, 7 moderate, and 1 severe. One of these AEs was related to the Prism software (e.g., headache) version 1, with two possibly related to the fMRI device. Most AEs were unrelated to the Prism device and included the following: sore throat, nausea, fever, cold, and abdominal pain. One patient had an SAE that manifested as abdominal pain, ultimately resulting in kidney stone removal (unrelated to the device or therapy).

### 3.2. Demographic and Baseline Characteristics (FA Set)

Table 2 shows the baseline characteristics of those enrolled in the trial. Table 3 shows the baseline values for the primary and secondary endpoints evaluated, with baseline HDRS-17 score 19.1 ± 6.26, and the baseline SHAPS-C score 38.8 ± 6.85, indicating moderate depression with significant anhedonia.

### 3.3. Clinical Outcomes (EF Set)

#### 3.3.1. Primary Efficacy Outcomes

The HDRS-17 primary endpoint, which is the change from baseline to the post-Prism NF training visit (6 weeks), is found in Table 4. The adjusted mean (LSmean) change from baseline for the HDRS-17 was −8.00 [95% CI: −10.5; −5.41]; *p* < 0.0001. The proportion of subjects who had a clinically meaningful reduction (at least a four-point reduction) from baseline in HDRS was 78.47% [95% CI: 58.83; 89.25%]. The proportion of patients achieving remission (HDRS ≤ 7) was 32.25% (11/34) [95% CI: 17.39%; 50.53%] (post hoc analysis). The proportion of subjects with at least a 50% reduction in the HDRS-17 score (responders) was 38.24% (13/34) [95% CI: 22.17%; 56.44%]. Based on the primary outcome results for the HDRS-17, the null hypothesis (H_0_: mean HDRS ≥ −4) was rejected.

#### 3.3.2. Secondary Efficacy Outcomes

The changes from baseline in the SHAPS-C, QIDS-SR-16, GAD-7, and PHQ-9 to 6 weeks (post training) for the EF set were found to demonstrate statistically significant improvements based on the repeated measures ANCOVA model described above. Notably, the proportion of subjects with at least a 50% reduction in the SHAPS-C score was 76.5% (26/34) [95% CI: 58.83%; 89.35%] (Table 4).

Figure 3 and Figure 4 show the average score for each time point as well as the change from baseline to the mid-point of the training and from baseline to end of training for HDRS-17 and SHAPS-C. Both HDRS-17 and SHAPS-the C scores showed progressive improvement from baseline through the mid-treatment and end-of-treatment assessments. There appeared to be a dose-dependent response to Prism training, which did not reach a plateau level after ten sessions.

#### 3.3.3. Effect Sizes and Clinical Significance

The effect sizes (standardized mean difference/Cohen’s d) evaluated from the baselines of the HDRS-17, SHAPS-C, QIDS-SR-16, GAD-7, and PHQ-9 scores to 6 weeks post training for the EF set were found to show statistically significant improvements and demonstrated moderate (0.5 to 0.8) to large (>0.8) effect sizes (Table 5).

#### 3.3.4. Patient Satisfaction

A five-item Patient Satisfaction questionnaire was used in this study, aiming to gauge subjective satisfaction with the therapy and conducted study. Table 6 presents the descriptive statistics of the responses per question for each of the analysis sets (1 = not satisfied through 5 = very satisfied). Overall, subject satisfaction with the Prism training was high, with 85.3% giving a score of 3 or higher out of a maximum score of 5.

## 4. Discussion

The pilot study of the Prism for Depression™ sessions in patients with anhedonic MDD demonstrated favorable outcomes across the following three key domains: (1) Safety: Only one device-related adverse event occurred (headache), which resolved without treatment. This favorable safety profile compares well with other biological and neuromodulation-based treatments, which often report higher rates of adverse events. (2) Efficacy: This self-neuromodulation intervention produced significant reductions in both depression and anhedonia with large effect sizes on clinician-rated measures (HDRS-17, SHAPS-C) and moderate improvements in self-reported measures of depression and anxiety (QIDS-SR-16, PHQ-9, and GAD-7); and (3) Acceptability: The majority of participants (>85%) reported satisfaction with the treatment, perceived it as effective, and would recommend it to others. This high completion rate (77%) compares favorably to the typically higher dropout rates seen with treatments like TMS and pharmacotherapy.

### 4.1. Addressing the Anhedonia Treatment Gap in MDD

As it relates to clinically meaningful change, this study met the threshold identified in the STAR*D report (4–6-point reduction) for HDRS-17, with an average reduction of 8.0 points (95% CI: −10.5 to −5.41, *p* < 0.0001), placing it in the range defined as “clinically substantial improvement”. Notably, 78.5% of patients achieved clinically meaningful improvement (≥4-point HDRS reduction) and 32.25% achieved remission (HDRS ≤ 7). The high proportion of patients showing clinical improvement suggests the intervention effectively addresses the core depressive symptoms in this population. The observed effect sizes were large for depression (HDRS-17, Cohen’s d = 1.22) and anhedonia (SHAPS-C, Cohen’s d = 0.92) measures, indicating substantial clinical impact that would be meaningful for both clinicians and patients.

The significant reduction in anhedonia symptoms (SHAPS-C average reduction of 6.3 points, 95% CI: −8.51 to −4.14, *p* < 0.0001) addresses a critical treatment gap. The improvement through VS upregulation further supports the use of the suggested mechanism for anhedonia of diminished reward processing [14]. Future research should delineate the subprocesses of anhedonia (i.e., incentive versus hedonic components), which represent distinct but related aspects of reward processing [14]. More precise biomarkers targeting specific regions within the reward circuit (e.g., different parts of the VS, medial vs. lateral, or ventromedial prefrontal cortex/orbitofrontal cortex connections) could substantially improve treatment personalization and outcome prediction [31]. Such precision would allow for tailoring of the intervention to address individual deficits in reward anticipation versus consumption.

Despite 66.6% of participants receiving concomitant SSRI/SNRI medications, these individuals showed robust decreases in depression and anhedonia, as measured by HDRS-17 and SHAPS-C. Indeed, SSRI/SNRI medications are known to have limited benefit for anhedonia and may even have pro-anhedonic effects in some individuals [30,32]. This suggests that RS-EFP-NF may address symptom dimensions not adequately targeted by conventional pharmacotherapy.

Notably, improvement demonstrated a progressive pattern from baseline through mid-treatment to end-of-treatment (Figure 3), suggesting a cumulative benefit pattern possibly pointing to the benefit of booster sessions. This temporal dynamic has important implications for treatment prediction and relapse prevention, though further controlled studies of dosage and frequency are necessary. Based on the data analysis of the mean, median, and mode (on the improvements from baseline to end of therapy) for the various instruments used, there does not appear to be a ceiling effect, suggesting that increasing the number of sessions or duration of sessions may have an additional positive effect on effectiveness.

### 4.2. Comparison with Existing Treatment Approaches

The remission rate of 32.25% compares favorably to typical ranges (11–30%) for effective depression treatments, as reported by Mendelewicz et al. (2008) [33]. Current pharmacotherapies, particularly SSRIs/SNRIs, often improve mood symptoms but leave residual anhedonia symptoms, representing a specific treatment challenge.

Conventional NF approaches for depression target either process/non-neuroanatomically defined EEG patterns (e.g., alpha asymmetry) or amygdala-fMRI-NF upregulation. [34] demonstrated the efficacy of real-time fMRI amygdala NF for MDD, focusing on emotion regulation with positive memories rather than reward processing. In contrast, RS-EFP’s focus on reward system activity offers both neuroanatomically precise and process targeted approach to anhedonia. As mentioned above, this can be further improved for subprocesses of reward processing through more granular neuroanatomical and or mental probing of reward anticipation versus consumption.

While NF has not been included in MDD treatment guidelines, recent systematic reviews and meta-analysis concluded that patients with depression showed significant cognitive, clinical, and neural improvements following electroencephalogram NF (EEG-NF or fMRI-NF) training [35,36]. The importance of self-driven treatment, enhancement of the sense of agency over one’s condition, and targeting endogenous mental processes represent unique benefits of NF compared to passive treatments. However, NF studies so far didn’t target the deep brain structures associated with anhedonia; a core and resistant symptom of MDD. 

The advantage of Prism is that it is a NF approach combining information from both EEG and fMRI through machine learning, making the technology affordable and specific to evaluating the functionality of the reward system which is meaningful for anhedonia. This aligns with the Research Domain Criteria [RDoC] framework [37], which encourages focusing on fundamental neurobiological dimensions across traditional diagnostic categories rather than symptom-based diagnoses alone. This process-based approach can fit into the treatment landscape as a complement to existing options, particularly for patients with inadequate response to standard treatments. 

### 4.3. Potential for Scalable Clinical Implementation/Feasibility

The RS-EFP-NF approach offers a non-invasive intervention that delivers a targeted approach to addressing reward system-related processes that underlie depression. Its potential for broader implementation stems from its minimal training requirements for both therapists and patients, the possibility of supervision by non-physician personnel, a well-defined treatment protocol with the potential for real-life skills translation (through mental strategy development), and a low adverse event profile. However, longer follow-up is needed to confirm the translation of learned skills to everyday situations.

Patient-centered self-regulation skills learned during treatment could potentially extend beyond the sessions. Practical advantages include accessibility and the potential for integration into existing treatment pathways, especially with psychotherapy. EFP-NF interventions have shown promising results when combined with traditional psychotherapeutic approaches in Post Traumatic Stress Disorder [38]. In treatment-resistant depression, this could be especially beneficial, considering that despite many existing pharmacological and psychological approaches, not everyone responds adequately.

### 4.4. Methodological Considerations and Limitations

This study has several important limitations that should be considered when interpreting the results. Most critically, the single-arm, open-label design without a control group substantially limits causal inferences. Without a sham NF control, placebo effects, expectancy, therapist attention, natural symptom fluctuation, or regression to the mean, cannot be adequately ruled out as explanations for the observed improvements.

The 23% attrition rate (10 of 44 enrolled participants not completing the protocol) raises concerns about selection bias and potentially limits generalizability to broader MDD populations. Those who completed the study may represent a subgroup more likely to respond to or tolerate the intervention. The sample characteristics (predominantly female, 75%; Caucasian, 93.2%; non-treatment-resistant) further limit generalizability to more diverse MDD populations.

The presence of concurrent medications in most participants (66% on SSRIs/SNRIs) represents a significant confound. The potential interactive effects between Prism and medications cannot be fully determined without specific analyses comparing outcomes between medicated and non-medicated participants.

The fixed treatment parameters (10 sessions) prevent determination of optimal dosing and boostering strategies. The absence of follow-up assessments beyond the end of treatment raises questions about the durability of the observed benefits, skill transfer, and relapse rates. Future studies should incorporate longer follow-up periods to assess the maintenance of gains.

We were unable to comprehensively assess the mechanisms of action or identify which specific aspects of reward processing (i.e., anticipation vs. consumption) were most affected by the intervention. More refined assessments of the reward processing subcomponent would be valuable for understanding the specific pathways through which RS-EFP-NF exerts its effects.

To address these limitations, a randomized controlled trial with appropriate blinding, sham-control, and longer follow-up is currently underway (NCT05869708).

## 5. Conclusions

The results of this pilot study suggest that RS-EFP-NF may represent a potentially useful approach to addressing anhedonia symptoms in MDD. The intervention demonstrated acceptable safety and was associated with improvements in both depression and anhedonia measures. However, the open-label design and lack of control group necessitate caution when interpreting these preliminary findings. If confirmed through controlled trials, this approach could represent a valuable addition to the therapeutic options for depression with anhedonia, particularly for patients with residual anhedonic symptoms despite standard treatments.

## Figures and Tables

**Figure 1 brainsci-15-00476-f001:**
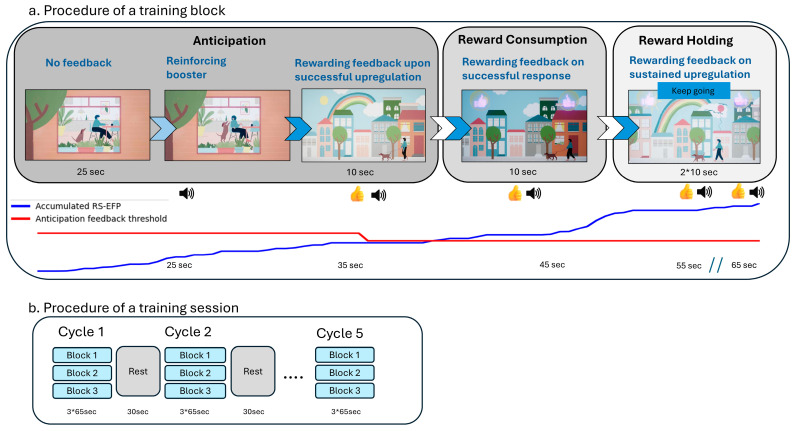
Prism for Depression™ design: (**a**) Feedback interface design and monitoring—Each training block progresses through Anticipation (25 s), Reward Consumption (10 s), and Reward Holding (20 s) phases. The blue line represents the accumulated RS-EFP signal, with the red line showing the anticipation threshold. (**b**) NF session protocol—Each training session consists of 5 cycles, with 3 blocks per cycle, separated by rest periods.

**Figure 2 brainsci-15-00476-f002:**
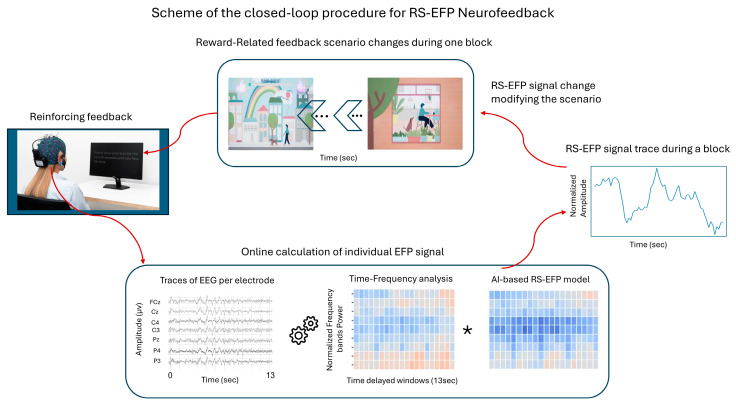
Schematic representation of the closed-loop RS-EFP-NF procedure. The system captures EEG signals from the trainee, processes them through a time–frequency analysis and an AI-based RS-EFP model, and generates real-time feedback that directly modifies the audio–visual scenario, creating a continuous learning loop for reward system activation (asterisk denotes multiplication).

**Figure 3 brainsci-15-00476-f003:**
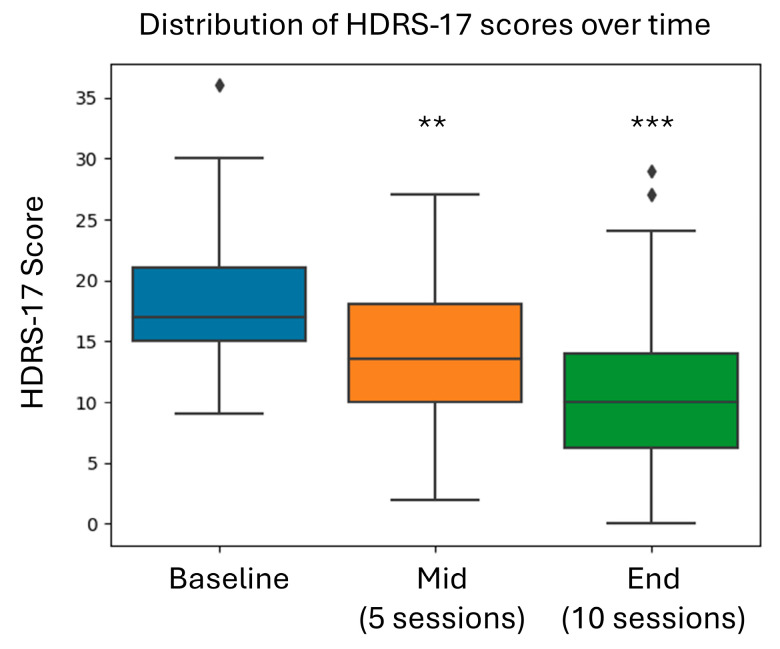
Change in the HDRS-17 score over time (mean ± SE). Asterisks indicate the statistical significance of the change from baseline. ** *p* < 0.0001 and *** *p* < 0.00001.

**Figure 4 brainsci-15-00476-f004:**
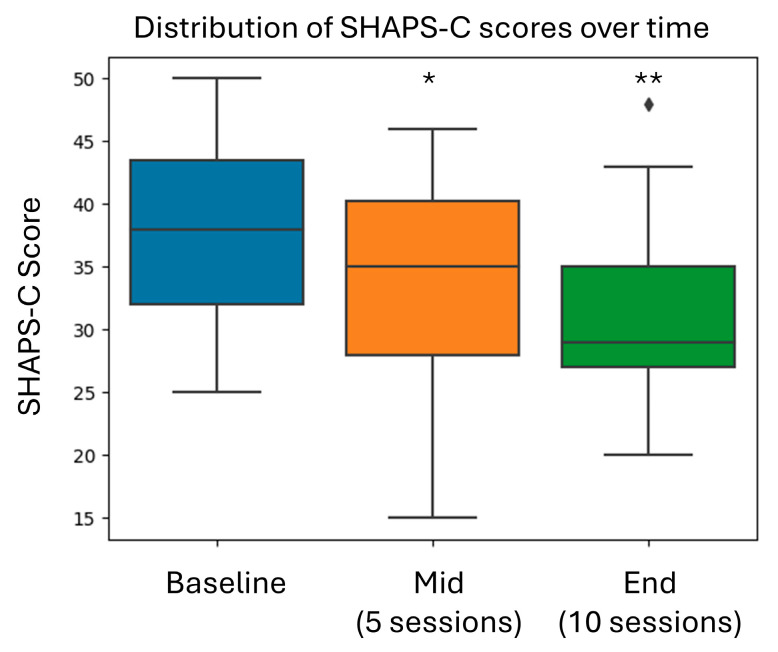
Change in the SHAPS-C score over time (mean ± SE). Asterisks indicate the statistical significance of the change from baseline. * *p =* 0.0032 and ** *p* < 0.0001.

**Table 1 brainsci-15-00476-t001:** The disposition of patients based on the FA, EF, and PP.

	Number of Subjects
Screened	49
Screen Failures	5
FA analysis set	44
Completed 1 session	1
Completed 3 sessions	2
Completed 4 sessions	1
Completed 6 sessions	1
Completed 7 sessions	2
Did not start NF sessions	3
EF analysis set	34
Major Protocol Violation	0
PP analysis set	34

FA = Full Analysis; EF = Efficacy Analysis; PP = Per Protocol; NF = Neurofeedback.

**Table 2 brainsci-15-00476-t002:** Demographic and baseline characteristics—FA set (N = 44).

			FA Set
Age (years)		N	44
		Mean (SD)	39.9 (11.03)
		Median [Range]	39.1 [21.5; 64.6]
Gender	Male	% (n/N)	25.0% (11/44)
	Female	% (n/N)	75.0% (33/44)
Ethnicity	Not of Hispanic or Latino origin	% (n/N)	100% (44/44)
Race	Caucasian	% (n/N)	93.2% (41/44)
	Other	% (n/N)	6.8% (3/44)
Years of Education	Did not finish high school	% (n/N)	2.3% (1/44)
	High school diploma or equivalent	% (n/N)	40.9% (18/44)
	Some college, no degree	% (n/N)	15.9% (7/44)
	Associate degree (for example: AA or AS)	% (n/N)	4.5% (2/44)
	Bachelor’s degree (for example: BA or BS)	% (n/N)	31.8% (14/44)
	Master’s degree (For example: MA or MS)	% (n/N)	4.5% (2/44)
Marital Status	Married	% (n/N)	40.9% (18/44)
	Divorced	% (n/N)	29.5% (13/44)
	Single	% (n/N)	29.5% (13/44)
Laterality	Right	% (n/N)	95.5% (42/44)
	Ambidextrous	% (n/N)	4.5% (2/44)
Duration of Current Episode		Mean (SD) in months	72 (88.8)
Concomitant Meds	SSRIs/SNRIs	% (n/N)	66% (29/44)
	Cannabis	% (n/N)	16% (7/44)
	Benzodiazepines	% (n/N)	30% (13/44)
	Other	% (n/N)	36.4% (16/44)
Comorbidities	Fibromyalgia	% (n/N)	20.5% (9/44)
	Insomnia	% (n/N)	6.8% (3/44)
	PTSD	% (n/N)	4.5% (2/44)
	Migraine	% (n/N)	4.5% (2/44)
	Irritable bowel syndrome (IBS)	% (n/N)	4.5% (2/44)

SD = standard deviation; AA = Associate of Arts; AS = Associate of Science; BA = Bachelor of Arts; BS = Bachelor of Science; MA = Master of Arts; MS = Master of Science; SSRI = selective serotonin reuptake inhibitor; SNRI = serotonin and norepinephrine reuptake inhibitor; PTSD = post-traumatic stress disorder.

**Table 3 brainsci-15-00476-t003:** Baseline values of the primary and secondary endpoints.

Primary and Secondary Endpoints	Baseline Value
HDRS-17	19.1 ± 6.26
SHAPS-C	38.8 ± 6.85
QIDS-SR-16	15.5 ± 5.03
GAD-7	11.4 ± 5.74
PHQ-9	12.4 ± 4.7

HDRS-17 = Hamilton Depression Rating Scale (17-item); SHAPS-C = Snaith–Hamilton Pleasure Scale, Clinician-administered; QIDS-SR-16 = Quick Inventory of Depressive Symptomatology, Self-Report; GAD-7 = Generalized Anxiety Disorder, 7-item scale; PHQ-9 = Patient Health Questionnaire, 9-item scale.

**Table 4 brainsci-15-00476-t004:** Changes from the baseline HDRS-17, SHAPS-C, CGI-I, QIDS-SR-16, GAD-7 and PHQ-9—EF set (mean difference).

Instrument	Baseline to 6-Week Assessment	
	LS Means (95% CI)	*p*-Value
HDRS-17	−8.0 (−10.5 to −5.41)	<0.0001
SHAPS-C	−6.3 (−8.51 to −4.14)	<0.0001
CGI-I	2.5 (2.22 to 0.72)	<0.0001
QIDS-SR-16	−4.3 (−5.97 to −2.62)	<0.0001
GAD-7	−3.3 (−4.47 to −2.12)	<0.0001
PHQ-9	−4.7 (−7.94 to −1.40)	<0.01

HDRS-17 = Hamilton Depression Rating Scale; SHAPS-C = Snaith–Hamilton Pleasure Scale, Clinician-administered; CGI-I = Clinical Global Impression-Improvement; QIDS-SR-16 = Quick Inventory of Depressive Symptomatology, Self-Report; GAD-7 = Generalized Anxiety Disorder, 7-item scale; PHQ-9 = Patient Health Questionnaire, 9-item scale; LS = least squares; CI = confidence interval.

**Table 5 brainsci-15-00476-t005:** Effect sizes from baseline to end of treatment (6 weeks).

Instrument	Effect Size	*p*-Value
HDRS-17	1.22 [95% CI: 0.7 to 1.74]	<0.00001
SHAPS-C	0.92 [95% CI: 0.42 to 1.42]	0.0003
QIDS-SR-16	0.81 [95% CI: 0.31 to 1.3]	0.001
GAD-7	0.54 [95% CI: 0.06 to 1.03]	0.03
PHQ-9	0.75 [95% CI: 0.03 to 1.47]	0.04

HDRS-17 = Hamilton Depression Rating Scale; SHAPS-C = Snaith–Hamilton Pleasure Scale, Clinician-administered; QIDS-SR-16 = Quick Inventory of Depressive Symptomatology, Self-Report; GAD-7 = Generalized Anxiety Disorder, 7-item scale; PHQ-9 = Patient Health Questionnaire, 9-item scale; CI = confidence interval.

**Table 6 brainsci-15-00476-t006:** Subject satisfaction with Prism.

		Mean (SD)
EF	To what extant were you satisfied with the NF training in this trial?	3.6 (1.05)
	In your opinion, how effective was the NF training?	3.6 (1.13)
	Would you recommend the use of the PRISM system (used in this trial) to your friends/family members?	3.7 (1.24)

Higher scores indicate greater satisfaction on a scale of 1–5.

## Data Availability

Data are available upon reasonable request from the corresponding author. The data are not publicly available for ethical and commercial reasons.

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
