# Peer review of "Reward System EEG–fMRI-Pattern Neurofeedback for Major Depressive Disorder with Anhedonia: A Multicenter Pilot Study"

_brainsci, 2025, doi:10.3390/brainsci15050476_

Round 1
Reviewer 1 Report
Comments and Suggestions for Authors
The manuscript presents a pilot study evaluating the safety and efficacy of a novel neurofeedback (device (Prism) targeting the reward system in MDD patients with anhedonia. While the study addresses an important gap in treatment for anhedonia, the manuscript suffers from significant methodological flaws, lack of rigor in statistical reporting, and overinterpretation of results.
Below I provide detailed critiques and recommendations for improvement.
- My biggest criticism of the study is that it doesn’t have a control so it’s not clear what should we compare the results to? The single-arm, open-label design severely limits causal inference. Without a sham or active control group, placebo effects, natural symptom fluctuation, or even nonspecific factors (e.g., therapist attention) can’t be ruled out. The authors acknowledge this but downplay its impact.
- There is high attrition. Only 34/49 subjects completed the protocol (30% attrition), raising concerns about selection bias and generalizability. The planned sample size (n=30) was arbitrary ("rule of thumb") and underpowered for detecting moderate effects.
- The mixed-model analysis lacks detail (e.g., random/fixed effects structure, covariance matrix). What authors says that "clinically substantial improvement" (HDRS-17 reduction >=7 points) is post-hoc and not pre-specified.
- Inclusion and exclusion criteria should be numbered 1, 2, 3.
- The RS-EFP biomarker’s specificity to reward processing is inadequately validated. The cited validation study (Singer et al., 2023) is not sufficiently described, and no data are provided to link RS-EFP changes to clinical improvement.
- 66% of participants were on SSRIs/SNRIs, which may confound results and this is a massive issue. No subgroup analysis was performed to assess Prism’s effects independent of pharmacotherapy.
- Given this is a multicenter study, the protocol’s usage across multiple sites introduces potential variability in administration.
- How long the results last? Durability of effects is unknown. Anhedonia often relapses and without follow-up data, the clinical relevance is limited.
- Conclusion should tone down. The manuscript claims Prism is "promising and safe" based on a small, uncontrolled pilot. Terms like "substantial clinical effects" are exaggerated without comparative data.
- Tables: all abbreviations should be mentioned in tables footnotes.
- Figures 3 and 4 should have labels removed from the figures itself.
- Reference list is limited and some important papers in the field are not cited.
- In text citation uses Roman numbers which make it very confusing to follow. The reference list uses both Roman and numbers.
Author Response
Reviewer 1
The manuscript presents a pilot study evaluating the safety and efficacy of a novel neurofeedback (device (Prism) targeting the reward system in MDD patients with anhedonia. While the study addresses an important gap in treatment for anhedonia, the manuscript suffers from significant methodological flaws, lack of rigor in statistical reporting, and overinterpretation of results.
Below I provide detailed critiques and recommendations for improvement.
- My biggest criticism of the study is that it doesn’t have a control so it’s not clear what should we compare the results to? The single-arm, open-label design severely limits causal inference. Without a sham or active control group, placebo effects, natural symptom fluctuation, or even nonspecific factors (e.g., therapist attention) can’t be ruled out. The authors acknowledge this but downplay its impact.
We agree that this is a significant limitation of the study. We have substantially expanded our discussion of this limitation in section 4.4 (Methodological Considerations and Limitations), explicitly acknowledging that without a control group, we cannot rule out placebo effects, expectancy, therapist attention, natural symptom fluctuation, or regression to the mean as explanations for the observed improvements. We have tempered our conclusions throughout the manuscript to reflect the preliminary nature of these findings and have noted that a randomized controlled trial is needed to address this limitation.
- There is high attrition. Only 34/49 subjects completed the protocol (30% attrition), raising concerns about selection bias and generalizability. The planned sample size (n=30) was arbitrary ("rule of thumb") and underpowered for detecting moderate effects.
We thank the reviewer for this observation. We would like to clarify that of the 49 screened participants, 44 were enrolled in the study, and 34 completed the protocol. This represents an attrition rate of 23% (10 of 44 enrolled participants) rather than 30%. We acknowledge this limitation in section 4.4, noting the potential for selection bias and the possibility that completers may represent a subgroup more likely to respond to or tolerate the intervention.
- The mixed-model analysis lacks detail (e.g., random/fixed effects structure, covariance matrix). What authors says that "clinically substantial improvement" (HDRS-17 reduction >=7 points) is post-hoc and not pre-specified.
We have revised the statistical methods section to provide more detail about the linear mixed model structure, including the fixed and random effects and the covariance structure. Specifically, we have clarified that the model included visit as a categorical factor, baseline HDRS-17 as a covariate, and site as a random effect with an unstructured covariance matrix to account for within-subject correlations. We have also preserved important information about our approach to multiple testing, site poolability assessment, and the statistical tests used for various comparisons. We have clarified that the primary endpoint was the mean change in HDRS-17 from baseline, with the performance goal of -4 points change.
- Inclusion and exclusion criteria should be numbered 1, 2, 3.
We have reformatted the inclusion and exclusion criteria as requested to improve readability and clarity.
- The RS-EFP biomarker’s specificity to reward processing is inadequately validated. The cited validation study (Singer et al., 2023) is not sufficiently described, and no data are provided to link RS-EFP changes to clinical improvement.
We have added more detailed information about the RS-EFP biomarker's development and validation, noting that it was developed through simultaneous EEG-fMRI recordings during reward processing tasks and validated in healthy individuals by Singer et al. [25], demonstrating its specificity for detecting reward system activation particularly in the ventral striatum. We have also added technical details about the electrode configuration (CZ, PZ, C3, C4, P3, and P4 (referenced to FCz)) used for capturing the EEG signals that predict VS activation.
- 66% of participants were on SSRIs/SNRIs, which may confound results and this is a massive issue. No subgroup analysis was performed to assess Prism’s effects independent of pharmacotherapy.
We acknowledge this important limitation in section 4.4. Given our relatively small sample size, a formal subgroup analysis comparing outcomes between medicated and non-medicated participants would be underpowered to detect meaningful differences. However, we have identified this as a significant confound that limits interpretation of the results and have emphasized that future studies with larger samples should include analyses comparing outcomes between medicated and non-medicated participants to better isolate the effects of the intervention.
- Given this is a multicenter study, the protocol’s usage across multiple sites introduces potential variability in administration.
We have clarified in the results section that there was no statistically significant difference between sites for the primary endpoint, which supports the pooling of data across sites.
- How long the results last? Durability of effects is unknown. Anhedonia often relapses and without follow-up data, the clinical relevance is limited.
We have acknowledged this limitation in section 4.4, noting that the absence of follow-up assessments beyond the end of treatment leaves questions about the durability of observed benefits. We have added that future studies should incorporate longer follow-up periods to assess maintenance of gains.
- Conclusion should tone down. The manuscript claims Prism is "promising and safe" based on a small, uncontrolled pilot. Terms like "substantial clinical effects" are exaggerated without comparative data.
We have significantly tempered the conclusion section to reflect the preliminary nature of the findings and the methodological limitations of the study. The revised conclusion emphasizes the need for caution when interpreting these findings and the importance of confirming them through more rigorous controlled trials.
- Tables: all abbreviations should be mentioned in tables footnotes.
We have added footnotes to all tables explaining the abbreviations used.
- Figures 3 and 4 should have labels removed from the figures itself.
We have revised the figure captions to be more comprehensive and will ensure that embedded labels are removed from the figures themselves in the final version.
- Reference list is limited and some important papers in the field are not cited.
We have expanded the reference list to include several important recent papers on neurofeedback and anhedonia in depression.
- In text citation uses Roman numbers which make it very confusing to follow. The reference list uses both Roman and numbers.
We have standardized all references to use Arabic numerals in sequential order throughout the manuscript and reference list.
Reviewer 2 Report
Comments and Suggestions for Authors
First of all, thank you so much for giving me this opportunity to review this manuscript.
Undoubtedly, the relevance of this problem is very urgent. Major depressive disorder (MDD) is diagnosed at an increasingly younger age each year. However, in 70% of cases of MDD, anhedonia is present, which worsens the quality of life of the patient and his family. For many patients with MDD, medication alone is not enough. In addition, increasing the dose of medication may worsen the disease or cause side effects.
The title matches the content.
The introduction is very interestingly presented and point by point. However, there is a lack of information about the presented device and how it works. For example, which EEG and fMRI changes are crucial for diagnosing MDD and for choosing a treatment program.
Please provide reference numbers in Arabic numerals.
The purpose is stated briefly, specifically, clearly and distinctly.
Please provide inclusion and exclusion criteria item by item.
Please provide the registration numbers and dates of approval of the documents of the ethics committees.
Add more information about sample size determination characteristics
Add figure with localization of 8 EEG electrodes.
Please list which EFP markers were used for feedback.
Please add information about fMRI and patterns which were used for feedback in the materials and methods.
Too long “statistical methods “section. If possible, shorten it.
The results are presented in great detail using appropriate statistical tools. The authors used tables and figures to highlight the obtained results and their reliable significance.
The discussion is presented in detail and point by point based on the obtained results with comparison of own results with the results of other authors. However, there is no information about the EEG and fMRI patterns used to determine the feedback in the treatment of patients with depression.
The conclusions are very brief and focus on the safety and effectiveness of the device used.
Overall, the topic is very interesting, but there are some unclear points about the mechanism of the device itself, especially about EEG and fMRI patterns. In addition, without a control group, it is difficult to judge the effectiveness of the device. It is necessary to exclude the placebo effect.
Author Response
Reviewer 2
First of all, thank you so much for giving me this opportunity to review this manuscript.
Undoubtedly, the relevance of this problem is very urgent. Major depressive disorder (MDD) is diagnosed at an increasingly younger age each year. However, in 70% of cases of MDD, anhedonia is present, which worsens the quality of life of the patient and his family. For many patients with MDD, medication alone is not enough. In addition, increasing the dose of medication may worsen the disease or cause side effects.
The title matches the content.
The introduction is very interestingly presented and point by point. However, there is a lack of information about the presented device and how it works. For example, which EEG and fMRI changes are crucial for diagnosing MDD and for choosing a treatment program.
We have added information about the RS-EFP biomarker's development and validation through simultaneous EEG-fMRI recordings, demonstrating its specificity for detecting reward system activation particularly in the ventral striatum.
Please provide reference numbers in Arabic numerals.
We have revised all references to use Arabic numerals in sequential order throughout the manuscript.
The purpose is stated briefly, specifically, clearly and distinctly.
Please provide inclusion and exclusion criteria item by item.
We have reformatted the inclusion and exclusion criteria into numbered lists for better clarity and readability.
Please provide the registration numbers and dates of approval of the documents of the ethics committees.
The information below has been shared with the editor.
Centre name and number |
Rambam |
Sheba |
Sourasky |
Barzilai |
Site Principal Investigator |
Eyal Fruchter, MD Haya Yaron, MD |
Raz Gross, MD |
Haggai Sharon, MD |
Daniela Amital, MD |
Ethics Committee |
Rambam Ethical Committee |
Sheba Ethical Committee |
Sourasky Ethical Committee |
Barzilai Ethical Committee |
Ethics Committee Address |
HaAliya HaShniya St 8, Haifa, Israel |
Derech Sheba 2, Ramat Gan, Israel |
6 Weizmann Street Tel-Aviv Israel, 64239 |
Ha-Histadrut St 2, Ashkelon, Israel |
Chairman |
Prof. Shimon Pollaek |
Prof. Dror Harats |
Prof. Shmuel Kivity |
Prof. Amos Katz |
Date of initial approval of the final protocol |
March 26th, 2023 |
May 2nd, 2022 |
February 22nd, 2022 |
March 9th, 2022 |
EC/IRB Approval No. |
0650-22-RMB |
9063-22-SMC |
0002-22-TLV |
0002-22-BRZ |
Study Initiation Visit Confirmation Letter |
Feb 22nd, 2023 |
June 8th, 2022 |
March 13th, 2022 |
May 30th, 2022 |
Date of approval of amendment 1 |
May 1st, 2023 |
Aug 16th, 2022 |
Aug 18th, 2022 |
Aug 8th, 2022 |
Date of approval of amendment 2 |
NA |
April 9th, 2023 |
April 15th, 2023 |
May 9th, 2023 |
The study was registered on the Israel Ministry of Health website (MOH_2022-02-22_010631).
Add more information about sample size determination characteristics
We have clarified that the planned sample size was based on a rule of thumb recommendation for pilot studies aimed at assessing safety and generating preliminary efficacy data.
Add figure with localization of 8 EEG electrodes.
We have added information about the 6-electrode EEG system used in the Prism device, including the specific electrode positions (CZ, PZ, C3, C4, P3, P4 (referenced to FCz)). We have also clarified that the RS-EFP model incorporates time-frequency EEG features across multiple frequency bands to predict ventral striatum BOLD activation during reward processing, allowing for real-time estimation of reward-related neural activity.
Please list which EFP markers were used for feedback.
We have added information about the EFP markers used for feedback based on the RS-EFP model.
Please add information about fMRI and patterns which were used for feedback in the materials and methods.
We have added information about how fMRI patterns were incorporated into the RS-EFP model that formed the basis for neurofeedback.
Too long “statistical methods “section. If possible, shorten it.
We have condensed the statistical methods section while retaining the essential information about the analyses performed.
The results are presented in great detail using appropriate statistical tools. The authors used tables and figures to highlight the obtained results and their reliable significance.
The discussion is presented in detail and point by point based on the obtained results with comparison of own results with the results of other authors. However, there is no information about the EEG and fMRI patterns used to determine the feedback in the treatment of patients with depression.
The conclusions are very brief and focus on the safety and effectiveness of the device used.
Overall, the topic is very interesting, but there are some unclear points about the mechanism of the device itself, especially about EEG and fMRI patterns. In addition, without a control group, it is difficult to judge the effectiveness of the device. It is necessary to exclude the placebo effect.
We have addressed both concerns by: (1) providing more details about the device mechanism based on the RS-EFP approach, and (2) explicitly acknowledging in the limitations section that without a control group, we cannot definitively rule out placebo effects. We agree that a randomized controlled trial with appropriate control conditions is necessary to establish the efficacy of this approach beyond placebo effects.
Round 2
Reviewer 1 Report
Comments and Suggestions for Authors
The authors have addressed my comments well.
Reviewer 2 Report
Comments and Suggestions for Authors
Thank you for the opportunity to review this important manuscript
The relevance of this topic is great. Since neurofeedback in the treatment of major depressive disorder with anhedonia has not been sufficiently studied.
The title is consistent with the content
In the revised version, the introduction is better and more emphasizes the main relevance of the topic.
The purpose is short and specific
Changes and additional information in the materials and methods have provided greater clarity in this section.
The results are presented in great detail using appropriate statistical tools. The authors used tables and figures to highlight the obtained results and their reliable significance.
The discussion is presented in detail and point by point based on the obtained results with comparison of own results with the results of other authors. Authors added more information about the EFP markers used for feedback based on the RS-EFP model and about how fMRI patterns were incorporated into the RS-EFP model that formed the basis for neurofeedback.
The conclusions in revised version are very brief and focus on the safety and effectiveness of the device used.